# Structure and mechanism of the alkane-oxidizing enzyme AlkB

Xue Guo[1,6], Jianxiu Zhang [ORCID][1,6], Lei Han[1,6], Juliet Lee[2,3], Shoshana C. Williams [ORCID][2,4], Allison Forsberg[2,5], Yan Xu[1], Rachel Narehood Austin [ORCID][2] ✉ & Liang Feng [ORCID][1] ✉

Alkanes are the most energy-rich form of carbon and are widely dispersed in the environment. Their transformation by microbes represents a key step in the global carbon cycle. Alkane monooxygenase (AlkB), a membrane-spanning metalloenzyme, converts straight chain alkanes to alcohols in the first step of the microbially-mediated degradation of alkanes, thereby playing a critical role in the global cycling of carbon and the bioremediation of oil. AlkB biodiversity is attributed to its ability to oxidize alkanes of various chain lengths, while individual AlkBs target a relatively narrow range. Mechanisms of substrate selectivity and catalytic activity remain elusive. Here we report the cryo-EM structure of AlkB, which provides a distinct architecture for membrane enzymes. Our structure and functional studies reveal an unexpected diiron center configuration and identify molecular determinants for substrate selectivity. These findings provide insight into the catalytic mechanism of AlkB and shed light on its function in alkane-degrading microorganisms.

Hydrocarbons are ubiquitous. Up to 800 million tons are released each year through a combination of natural seeps, accidental releases from the petroleum industry, and the biological activity of cyanobacteria[1–3]. These molecules are a rich food source for hydrocarbon-metabolizing bacteria and have major impacts on ecosystems[4,5]. Accordingly, an indispensable step in the carbon cycle is the enzymatic transformation of liquid alkanes to render them biologically useful.

Bacteria that can metabolize liquid alkanes are found worldwide, from the equatorial region[6,7] to oil-impacted gulfs[8] to hydrocarbon-rich, pristine soils in the Arctic and Antarctica[7,9–22]. Across these environments, the primary enzyme that catalyzes the oxidation of liquid alkanes is alkane monooxygenase (AlkB). AlkB oxidizes a range of straight-chain alkanes and is widespread in diverse bacteria that use alkanes as their sole carbon and energy source[10,23–37]. Indeed, it was the most differentially expressed gene in the microbial community that grew rapidly after the Deepwater Horizon oil spill[38].

AlkB belongs to the membrane fatty acid desaturase (FADS)-like superfamily, which represents a group of non-heme diiron monooxygenases that desaturate or hydroxylate fatty acyl aliphatic chains[39]. Reflecting the catalytic plasticity of the membrane FADS-like superfamily, AlkB selectively hydroxylates the terminal methyl group of straight-chain alkanes; other members of the family introduce either double bonds or hydroxyl groups into lipid-based substrates, respectively. The first glimpse into the structure of these unique biocatalysts was provided by structures of mammalian stearoyl-CoA desaturase SCD1 and yeast sphingolipid α-hydroxylase Scs7p[40–43]. Except for the signature histidine-rich motifs, AlkB does not share significant sequence similarities or electron transfer partners with those two fatty acid enzyme families. The exact configuration of AlkB's diiron center remains unclear. Functional AlkBs have also been identified in pathogenic genera, such as *Mycobacterium tuberculosis* H37RV, *Legionella pneumophilia strain Philadelphia*, and *Pseudomonas aeruginosa* PAO1[31,36,44], which suggests AlkB may fuel pathogen growth[36]. More

[1]Department of Molecular and Cellular Physiology, Stanford University School of Medicine, Stanford, CA 94305, USA. [2]Department of Chemistry, Barnard College, 3009 Broadway, New York, NY 10027, USA. [3]Present address: Department of Biochemistry and Molecular Biophysics, California Institute of Technology, Pasadena, CA 91125, USA. [4]Present address: Department of Chemistry, Stanford University, Stanford, CA 94305, USA. [5]Present address: Department of Chemistry, University of Southern California, Los Angeles, CA 90007, USA. [6]These authors contributed equally: Xue Guo, Jianxiu Zhang, Lei Han. ✉e-mail: raustin@barnard.edu; liangf@stanford.edu

than 20,000 AlkB protein sequences have been identified, reflecting its widespread distribution across microorganisms.

Despite its fundamental importance to the global carbon cycle and to human health, AlkB's catalytic mechanism remains mysterious, and the enzyme's structure is yet to be experimentally determined. Central mechanistic questions remain unanswered: what features differentiate AlkB from the fatty-acid enzymes and enable it to selectively hydroxylate the terminal methyl group of linear alkanes; what features differentiate one AlkB from another, which, in turn, endow the entire family with the remarkably wide substrate range that makes it a cornerstone in the global carbon cycle. The paucity of information has left a substantial gap in our understanding of how AlkB's structure facilitates its function and has greatly hindered its application to biotechnology.

AlkB is one of a relatively small number of enzymes whose established function is to transform alkanes into alcohols under ambient conditions (Fig. 1a). This is a chemically difficult reaction because of the non-polar nature of the C-H bond and its large dissociation energy, as well as the challenges associated with the selective activation of $O_2$. Other alkane-oxidizing enzymes include particulate copper-containing methane monooxygenases (pMMO), heme-containing cytochrome P450 enzymes, two flavin-dependent enzymes LadA and AlmA[45], and another non-heme diiron enzyme, soluble methane monooxygenase (sMMO). AlkB is unique among alkane-oxidizing enzymes because it features a nitrogen-rich ligand environment surrounding the metal center; the other non-heme diiron enzymes have carboxylate-rich iron centers. This suggests AlkB has a distinct catalytic mechanism. Since decades of functional studies have been unsuccessful in defining this mechanism, we sought to gain structural insights into its active site.

Here, we report a cryo-EM structure of AlkB and concomitant characterizations of its catalytic activities. Our study reveals the architecture of this family of enzyme, key features of the diiron center, and molecular determinants of substrate selectivity, which help reconcile >50 years of data on this important but elusive driver of the global carbon cycle.

## Results

### Structural determination

To facilitate structural studies on AlkB, we systematically characterized the biochemical behaviors of a large panel of AlkB homologs. *Fontimonas thermophila* AlkB (FtAlkB), which shares 62% sequence identity with the prototype *Pseudomonas oleovorans* AlkB (GPo1AlkB, Supplementary Fig. 1), was found promising in terms of protein yield and stability (Supplementary Fig. 2a). FtAlkB displayed robust alkane hydroxylase activities on a model alkane octane (5 µmoles product/mg protein), which compares favorably to the most active AlkB we had previously characterized (4.5 µmoles product/mg protein for *Alcanivorax borkumensis* AlkB[46]) (Fig. 1b). These attributes marked FtAlkB as an excellent candidate for structural and mechanistic studies.

Since the catalytic activity of AlkB depends on iron, we measured FtAlkB's iron content using inductively coupled plasma mass spectrometry (ICP-MS). Our results showed that the purified FtAlkB is loaded with iron, while we did not detect any iron above the background in a control protein. In addition, we did not detect zinc

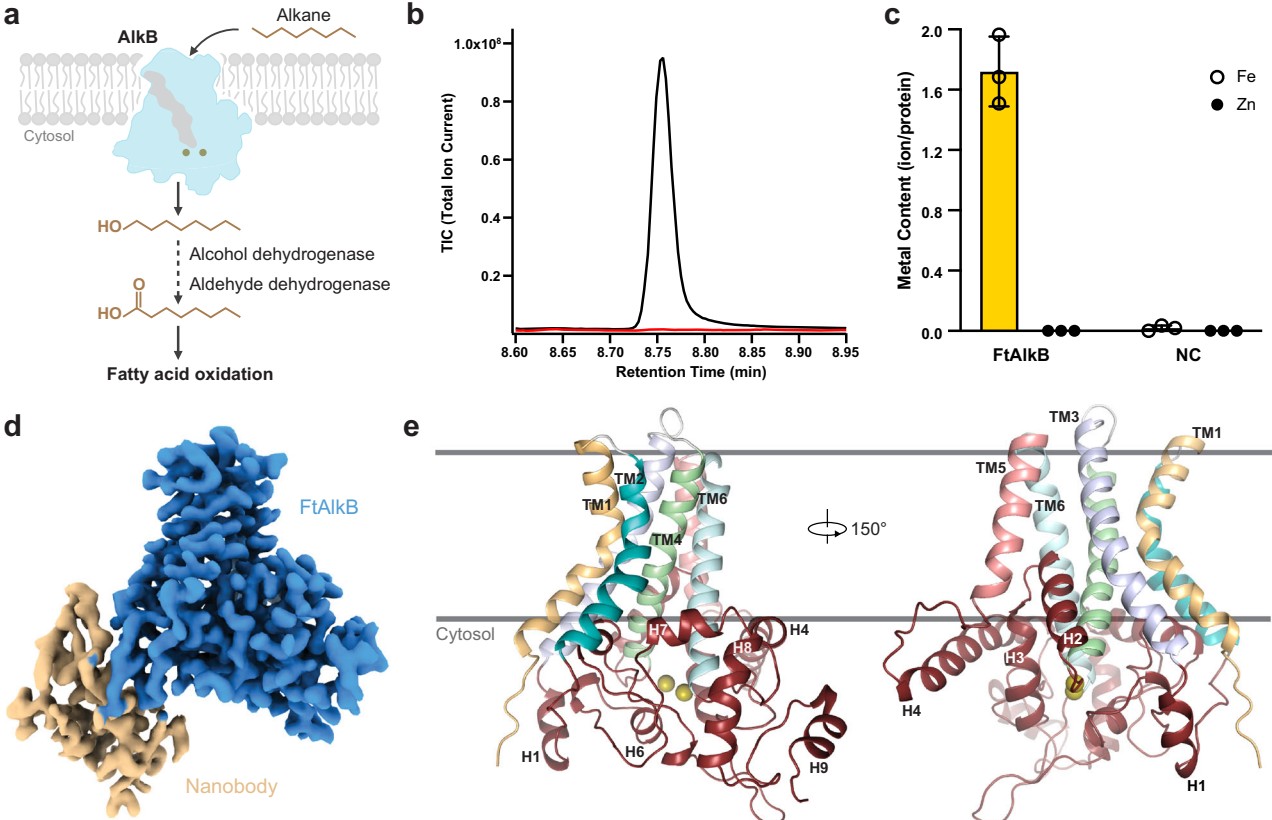

**Fig. 1 | Functional characterization and structure of FtAlkB. a** Key steps in the biological degradation of alkanes. AlkBs initiate alkane metabolism by transforming alkanes into alcohols. **b** Alkane hydroxylation activity of FtAlkB on octane. A representative GC-MS profile of the octanol peak of FtAlkB (black) and negative control without FtAlkB (red). Experiments were repeated three times independently and similar results were observed. The activity of the purified protein corresponds to 225 turnover numbers or 5 µmoles of product per mg of protein. Source data are provided as a Source Data file. **c** Iron occupancy of FtAlkB determined by ICP-MS (NC, negative control; mean ± SEM; $n = 3$ independent experiments). Source data are provided as a Source Data file. **d** Cryo-EM density map of FtAlkB-nanobody complex. **e** Overall structure of FtAlkB, viewed parallel to the membrane. Two iron atoms are shown as olive spheres.

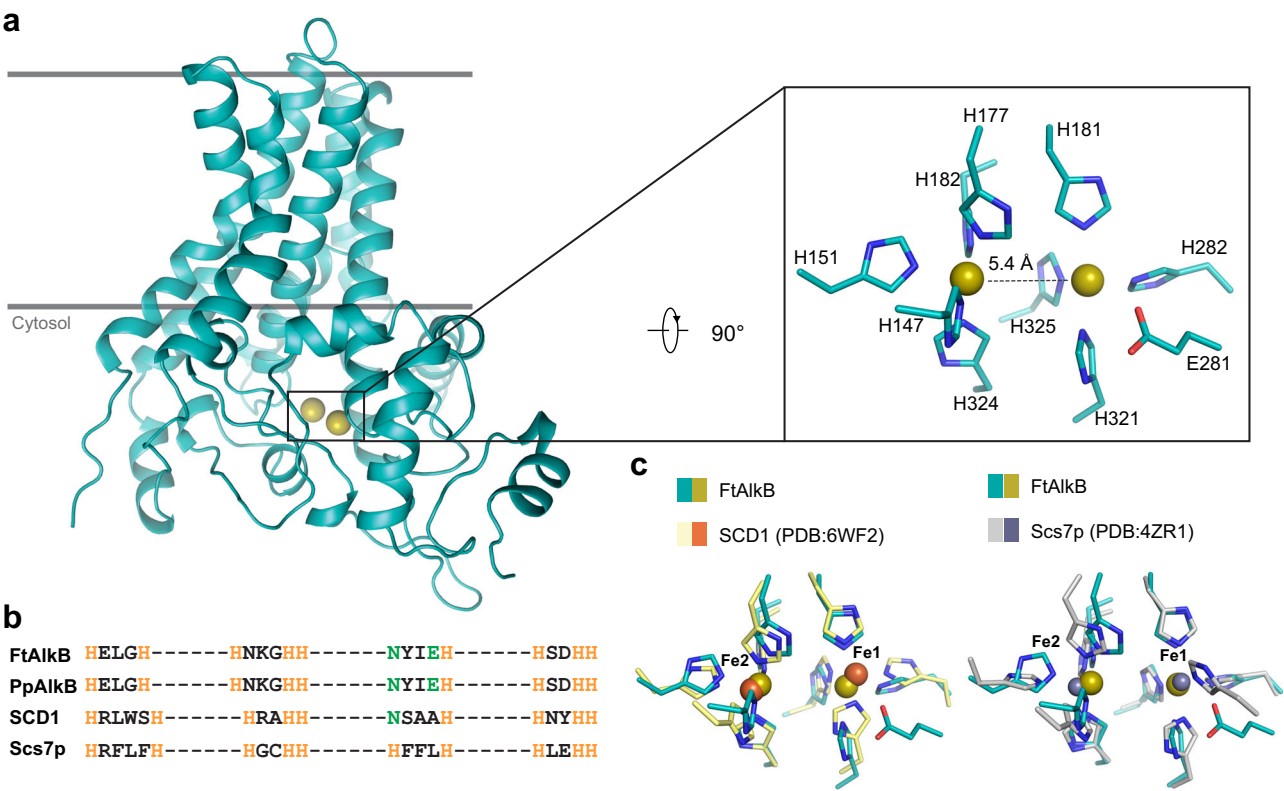

**Fig. 2 | The diiron center of FtAlkB. a** Structure of FtAlkB with a close-up view of the diiron coordination. The coordinating histidine residues and a surrounding glutamate are shown in sticks. **b** Alignments of conserved histidine motifs. **c** Superimposition of diiron centers in the members of the FADS-like superfamily.

above the background level (Fig. 1c). These results show that FtAlkB specifically binds iron. The specific iron content is roughly two iron ions per protein molecule, in line with the structural observations we present below.

To gain insights into the molecular mechanisms of the AlkB-catalyzed reaction, we carried out single-particle cryo-EM studies on FtAlkB purified in the detergent *n*-Dodecyl-β-D-Maltopyranoside (DDM, Supplementary Fig. 3 and Supplementary Table 1). To overcome the protein's relatively small size (45 kD) and to facilitate particle alignment, we developed a nanobody that specifically binds to FtAlkB and forms a stable complex (Supplementary Fig. 2b). With the aid of a nanobody as a fiducial marker, we obtained 3D reconstructions of FtAlkB at a resolution of 3.45 Å (with a mask on FtAlkB only; 3.59 Å with a mask on FtAlkB-nanobody), and the local resolution of FtAlkB reaches 2.8 Å to 3.2 Å. The high-quality map of FtAlkB allowed us to unambiguously trace the protein and build a model (Fig. 1d, e and Supplementary Fig. 4a).

### Overall architecture

The FtAlkB structure consists of a transmembrane domain (TMD) and a catalytic domain. The 'positive-inside' rule[47] predicts that both N- and C-termini of FtAlkB should reside on the intracellular side of the membrane, placing the catalytic domain intracellularly. This is consistent with the intracellular location of the redox partners that presumably interact near the catalytic domain. In the TMD, six TMs are arranged in a tent-like shape with their cytosolic ends apart from each other, creating a large interface to accommodate the catalytic domain (Fig. 1e). In addition, the H2 and H3 helices form a reentry loop that is inserted halfway into the membrane. At the membrane-cytosol interface, three helices parallel to the membrane surface help anchor the cytosolic domain onto the membrane and position the structural elements to support the formation of the active site. The cytoplasmic ends of TM4 and TM6 protrude out of the membrane and provide two histidine-rich motifs that coordinate the two iron ions. The diiron site

is further enclosed by two cytoplasmic segments where the other two histidine-rich motifs reside: a helix-turn-helix between TM4 and TM5, and a long C-terminus with short α-helices (Supplementary Fig. 4g).

Based on a search of the Protein Data Bank using Dali server[48], FtAlkB does not share significant structural similarity to other proteins with experimental structures. AlkB represents a fold that differs substantially from other known protein families, including those in FADS-like superfamily, such as fatty acid desaturase SCD1 and sphingolipid α-hydroxylase Scs7p (Supplementary Fig. 4e–g).

### The catalytic active site

In FtAlkB, two iron ions are coordinated by nine histidine residues from four conserved histidine-rich motifs (Fig. 2a and Supplementary Fig. 1). One iron is coordinated by five imidazole rings while the other is bound by four (Supplementary Fig. 4b). These histidine residues are invariant across species and were found to be indispensable for the enzymatic functions in GPo1AlkB[39,49]. The coordination geometry for both irons exhibits some characteristics of an octahedral arrangement, with the other iron appearing along the axis where a ligand is missing; this leaves the two iron ions separated by ~5.4 Å. Our structure directly depicts a nitrogen-rich environment of the diiron center in AlkB. This corroborates the identity of metal ligands surmised from previous spectroscopic studies of AlkB[50], and it contrasts sharply with the oxygen-rich coordination found in other known non-heme diiron alkane monooxygenases. We also observed fundamental differences in the configuration of the diiron centers between AlkB and other diiron alkane monooxygenases, such as sMMO. First, the distance between the two ions in FtAlkB is longer at ~5.4 Å compared to the iron-iron distance 3 Å in sMMO. Second, no bridging ligand is observed for FtAlkB, whereas sMMO has carboxylate and hydroxide groups bridging two iron ions. These key differences indicate that AlkB may use a previously unreported mechanism for alkane oxygenation.

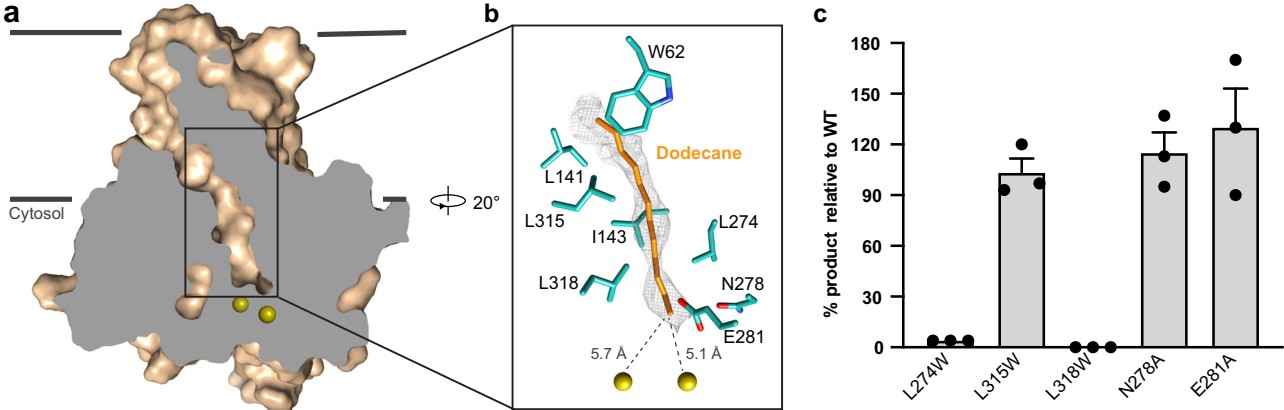

**Fig. 3 | Substrate channel of FtAlkB. a** A slice view of the substrate channel of FtAlkB. Two iron atoms are shown as olive spheres. **b** A close-up view of the ligand density fitted with dodecane. Key cavity residues are shown in sticks. **c** Alkane hydroxylation activities of FtAlkB mutants on octane, normalized to wild type (mean ± SEM; $n = 3$ independent experiments). Source data are provided as a Source Data file.

This di-metal configuration of AlkB is supported by observations in other lipid-metabolizing proteins in FADS-like superfamily, including fatty acid desaturases and sphingolipid α-hydroxylases. Although AlkB shares limited overall sequence and structural similarity with them, AlkB's signature histidine-rich motifs are organized similarly in space, and iron-coordinating histidine residues form a diiron binding site that resembles those in fatty acid desaturases and sphingolipid α-hydroxylases (Fig. 2b, c). The varying spacing between histidine residues is compensated by distinct helical conformations. These observations suggest a certain degree of shared catalytic mechanism among these nitrogen-rich diiron centers.

Nevertheless, there are important differences: there is a carboxyl group of a conserved glutamate residue, E281, in AlkB that is near Fe1 (the iron coordinated by four histidine residues) that could potentially interact with that iron and nearby histidine residues, although the side chain of E281 is not sufficiently resolved in the cryo-EM map. In contrast, a water molecule occupies a similar position in SCD1 while Scs7p has one additional histidine forming five histidine coordination for both irons (Fig. 2c). These differences in irons' surrounding environment may affect the reactivity of irons and the stability of reaction intermediates, which may play a role in the distinct reactions catalyzed by these enzymes.

**Substrate-binding vestibule**
Understanding how inert hydrophobic substrates reach AlkB's active site is critical to elucidating its hydrocarbon processing ability. TM2, TM4, and TM6 form an elongated vestibule that extends from the TM domain into the catalytic domain and reaches just above the active site. The vestibule is lined mostly by hydrophobic residues, and the vestibule's diameter is compatible with alkane binding (Fig. 3a). Interestingly, we found an elongated tubular non-protein density occupying the vestibule (Fig. 3b and Supplementary Fig. 4c). One end of the tubular density is near the active site and the distal end is close to W62. The shape of the density is compatible with an acyl chain, and the surrounding hydrophobic residues could help to accommodate an alkane. We tentatively placed dodecane into the density based on the length match, although the density's molecular identity cannot be unambiguously resolved at this resolution. The end of the alkane is positioned at a similar distance to the two irons (-5.1 Å and -5.7 Å, respectively; Supplementary Fig. 4d).

Among vestibule-lining residues, three leucine residues, L274, L315, and L318, are highly conserved (Supplementary Fig. 1). L274 and L318 interact with alkane near its terminus (the site of oxidation) from the opposite side. These two residues thus help position the alkane

properly for the catalytic reaction. L315, located at the distal part of the hydrophobic cavity, also contacts the alkane (Fig. 3b). In addition, the conserved $L^{315}X_2L^{318}$ motif links a hydrophobic helix (H4) to the last histidine-rich motif ($H^{321}X_2HH^{325}$). In our functional studies, substituting tryptophan for L274 or L318 abolished FtAlkB's catalytic activity (Fig. 3c). On the other hand, L315W substitution had little effect on the activity. These results highlight the importance of sidechain size for L274 and L318, consistent with their proposed role in positioning the substrate for the active site. The distal region of the substrate channel, where L315 is located, may accommodate more variability during the catalysis.

**Substrate selectivity**
In the substrate vestibule, W62 sits at a strategic position, packing against the last segment of the alkane. W62 and L141 (which interact with the alkane's terminus from the other side) define a narrow portion of the vestibule. This narrow region may present a hurdle to accommodating alkanes with more than 12 carbons. Notably, previous studies showed that the equivalent position of W62 in GPo1AlkB (W55) is correlated with the differential ability of bacteria to grow on alkanes: GPo1AlkB and its close homologs with tryptophan in this position can grow on alkanes with 5 to 12 carbons, with the optimal alkane chain length being close to 8. In contrast, AlkBs that possess a small hydrophobic residue (A, V, L, or I) at the same position can metabolize longer alkanes, specifically those with at least 12 carbons[44,49,51]. These observations are consistent with an important role for W62.

To further test the hypothesis that the residues that restrict the substrate channel dictate substrate selectivity, we assessed the activity of FtAlkB and its W62V variant on alkanes ranging from 5–14 carbons (Fig. 4a, b and Supplementary Fig. 5a). FtAlkB is active on alkanes with 6–12 carbons, with optimal activity on heptane and octane. For alkanes with >12 carbons, we observed no detectable activity. The W62V variant clearly shifted the substrate selectivity: it showed significantly increased relative activities for longer-chain alkanes such as decane, undecane, and dodecane. The W62V mutant can also oxidize tridecane and tetradecane. Our experiments using GPo1AlkB showed a similar trend (Supplementary Fig. 5b). Interestingly, a recent mutagenesis study on *D. cinnamea* AlkB showed that changing a less bulky amino acid at the position equivalent to FtAlkB's W62 into a bulkier one (V91W) decreased activities on longer alkanes[52]. These results underscore the important role the sidechain size of this position plays in determining the length of optimal substrates.

All of these results together are consistent with a model in which each AlkB has evolved a substrate channel that closely matches the size

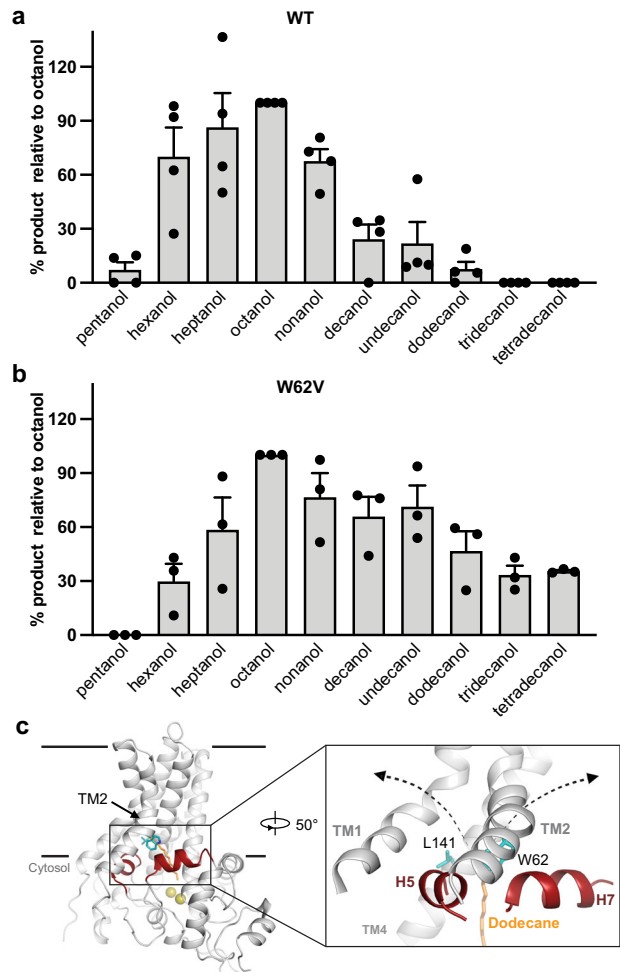

**Fig. 4 | W62V shifts the substrate length of FtAlkB. a, b** Alkane hydroxylation activities of WT (**a**) and W62V (**b**) on straight alkanes ranging from 5 to 14 carbons (mean ± SEM; $n = 4$ independent experiments for WT, $n = 3$ independent experiments for W62V). Source data are provided as a Source Data file. **c** Potential lateral pathways for substrate entry or product release. Dodecane is shown in orange. Key residues are shown in cyan sticks and selected short helices are shown in red.

of its target alkanes, much like a long evening glove tightly fits the hand and arm that slides into it.

## Discussion

Our structure of FtAlkB reveals the architecture of this family of important enzymes. AlkB adopts a protein fold not previously observed experimentally, while sharing an overall conserved di-metal coordination geometry with membrane fatty acid desaturases and sphingolipid α-hydroxylases. The AlphaFold[53] model (AF-A0A1I2KHB9-F1) aligns well with our FtAlkB structure (root mean square deviation (r.m.s.d) = 1.163 Å, Supplementary Fig. 4e), demonstrating its ability in predicting an unknown structural fold. On the other hand, the lack of metal ions in the AlphaFold predicted structure limits its catalytic insights.

Our structure and functional studies provide important insights into how AlkB achieves substrate selectivity. The substrate's terminal methyl group is positioned near the diiron active site. A bulky amino acid ~13 Å from the active site differentiates AlkBs that preferentially oxidize alkanes of approximately eight carbons from those that efficiently oxidize longer alkanes (and concomitantly less efficiently oxidize shorter alkanes[54]), which have a less bulky residue in that position. It is conceivable that the bulky amino acid may serve as a barrier to

longer alkanes while also helping to hold optimally sized non-polar alkanes in place[54]. Potentially, other structural features in the substrate channel might provide additional discrimination. Our structure thus provides a foundation for future efforts to engineer AlkB for biotechnology applications.

In our AlkB structure, TM2, which anchors the distal end of the substrate, does not interact with adjacent TMs on the cytosolic side, leaving lateral openings to the lipid bilayer. Each opening is guarded by a hydrophobic helix parallel to the membrane, suggesting a possible gate that allows substrate entrance or product release (Fig. 4c). This suggests TM2 might control a lateral pathway that allows substrates to enter the channel and products to exit it. Such a lateral gate would allow alkanes that partition into the membrane to reach the active site via the shape-matched substrate channels of AlkB. Similarly, after the reaction, the alcohol products would be released through the same pathway so they can be further oxidized via the fatty acid oxidation system to provide energy and a carbon source to the bacterium.

Selective C-H bond activation is a holy grail of chemistry[55]. Thus, how members of the FADS-like superfamily do C-H activation chemistry has long been a matter of intense interest. Our structure provides valuable insights into this process. In our structure, AlkB positions the terminal methyl group near the diiron active site. For the fatty acid desaturases, internal C-C bonds are positioned to be desaturated. These observations together suggest that members of the FADS-like superfamily position long alkyl chain substrates in a substrate channel in such a way that the bonds to be broken are adjacent to the active site. Catalysis is thought to create a common high-valent intermediate that can bifurcate to desaturate or hydroxylate.

We and others had hypothesized that the FADS-like superfamily generates a catalytically competent intermediate that is similar to compound Q of the other well-characterized diiron C-H activating enzyme, sMMO[54,56-59]. It was thought that two iron ions of AlkB would be electronically connected to provide an energetic advantage in sharing the electron-deficient intermediates that are required to abstract a hydrogen atom from a strong C-H bond. Unexpectedly, our structure shows AlkB's nitrogen-rich active site has a long distance between the two iron ions and lacks a bridging ligand. This configuration is fundamentally different from the oxygen-rich non-heme diiron center of sMMO. Instead, AlkB's active site resembles that of membrane fatty acid desaturases and sphingolipid α-hydroxylases (irrespective of bound iron or incidentally incorporated zinc ion in those two enzymes). These observed comparable active site configurations from multiple distinct enzymes under different conditions strongly suggest that the diiron center formed by histidine residues in the FADS-like superfamily differs sufficiently from that of sMMO to require a distinct catalytic intermediate.

The structure presented here, combined with prior mechanistic work from our group, enables us to speculate on AlkB's mechanism[46,54,58-61] (Supplementary Fig. 6). The long-distance observed between the two iron ions suggests a mechanism in which the two irons play different roles, with one iron serving as an electron relay and a site for water binding, while the other iron ion serves as the site of catalysis. Proton-coupled electron transfer (PCET) from a Fe(II)-OH$_2$ species on iron one to the second iron ion where O$_2$ is bound, would form a ferric peroxo species and a ferric hydroxide species, which would be readily protonated. Heterolytic cleavage of the O-O bond would form a ferric hydroxo species on one iron, water, and the catalytically competent Fe(V) oxo on the other iron. Supporting this hypothesis is the fact that the one other metalloenzyme with a substrate radical lifetime comparably long to AlkB is naphthalene dioxygenase (NDO), where a formally Fe(V) oxo species is postulated to be the active species[62]. As has been seen with Mn(V) oxo species, the electronic structure of the rebounding species (in the case of AlkB, Fe(IV)-OH) impacts the radical lifetime[63]. It also remains possible that there might be a substantial rearrangement of the active site during

the reaction, not seen in our structure, which could lead to a more sMMO-like reaction mechanism[64]. Detailed spectroscopic characterization of reaction intermediates will be the subject of future work.

## Methods

### Protein expression and purification

DNA encoding FtAlkB (amino acids 4–399) was codon optimized and cloned into pRSF vector (Novagen). The protein was overexpressed in *E. coli* BL21 (DE3). Cells were grown at 37 °C and induced by 0.2 mM IPTG in the presence of 10 mg/L $Fe^{3+}$ at 30 °C for 5 h. Pelleted cells were resuspended in lysis buffer (20 mM HEPES pH 7.4, 150 mM NaCl, 20 mM imidazole) and disrupted by sonication. FtAlkB was solubilized by 1% DDM (Anatrace) at 4 °C for 1 h and then isolated from clarified supernatant by cobalt resin. After overnight digestion with 3 C protease, FtAlkB eluent was incubated with nanobody AP32 at a 1:2 molar ratio. The complex was purified by size-exclusion chromatography (SEC) on a Superdex 200 increase column (GE Healthcare) in buffer containing 20 mM HEPES pH 7.4, 150 mM NaCl, and 0.02% DDM and subjected to cryo-EM analysis.

### Alkane hydroxylation activity assay

Cells expressing AlkB were centrifuged and resuspended in lysis buffer and broken open by a single high-pressure pass in the French press and then centrifuged at $7000 \times g$ for 20 min. To 0.5 mL cell supernatant was added 0.5 mL cell supernatant from cells expressing AlkG and AlkT and 2 μL purified substrate. The reaction was initiated by adding sufficient NADH to achieve a working concentration of 12.3 mM. Assays were incubated while shaking at 37 °C for 30 min. At the end of the incubation period, the reaction was quenched with 100 μL $CHCl_3$. The mixture was vortexed for 1 min, then centrifuged at $16,200 \times g$ for 5 min. The organic layer was transferred and injected into Agilent 7820 A Gas Chromatography with an Agilent 5977E Mass Selective Detector (MSD). 1 μL sample was injected into an Agilent HP-5 column with an injector temperature of 275 °C and a split ratio of 25:1. The initial oven temperature was 45 °C, with a hold time of 2.25 min. The oven temperature was then increased at a rate of 10 °C per min up to a final temperature of 225 °C. The identity of each compound was determined by manually comparing the fragmentation pattern of each peak to that of an authentic standard, injected under identical conditions, and eluted at the same time. Peak intensities were determined by manual integration of the total ion current (TIC) using MSD ChemStation F.01.03.2357. All activity assays included a control with substrate + buffer and a control with AlkG, AlkT, NADH, and substrate. Purified proteins were assayed in a similar manner except that purified AlkG, maize ferredoxin reductase, and NADPH were used[46,65].

### ICP-MS

Purified proteins were digested overnight in 50% $HNO_3$ and then boiled for 30 min. Samples were diluted with milli-Q water to 5% $HNO_3$ and subjected to ICP-MS spectrometer (Thermo Scientific XSERIES 2). AlkB proteins and negative control, non-iron-containing semiSWEET transporter, were prepared in parallel. Iron standards were used for calibration.

### Anti-FtAlkB-nanobody selection and purification

FtAlkB-specific nanobody was generated from a synthetic nanobody library following published protocols[66,67]. After three runs of selection, individual nanobody candidate was sequenced and expressed in *E. coli* BL21 (DE3). Purified nanobodies were validated by their binding ability to FtAlkB through pull-down assay and analytical SEC. Nanobody AP32 was identified to form a stable complex with FtAlkB. The complex was subjected to cryo-EM studies.

### Cryo-EM sample preparation and data collection

For sample preparation, 3 μL of concentrated FtAlkB (19 mg/ml) was incubated on freshly glow-discharged holey carbon grids (Quantifoil Au R1.2/1.3 300 mesh) for 5 s, blotted (time 3, force 3) and then cryo-cooled in liquid ethane on a Vitrobot Mark IV (Thermo Fisher Scientific) at 4 °C under 100% humidity. Grids were screened on a 300 kV Titan Krios with Falcon 4 detector and Selectris energy filter (slit width 10 eV). During 8.14 s exposure time, 50 movie frames were collected by EPU software (Thermo Fisher Scientific) at a physical pixel size of 0.95 Å with a defocus of 1.6 μm to 2.2 μm and an electron exposure dose of 50 electrons per $Å^2$.

### Cryo-EM data processing

A total of 5072 movies was motion-corrected by MotionCorr2[68], yielding dose-weighted images and summed images. The dose-weighted images were imported and processed in cryoSPARC v3.3[69]. The contrast transfer function (CTF) of images was estimated by Patch CTF estimation. Images with a CTF resolution better than 8 Å were selected, yielding 4,965 images. A half dataset was used to auto-pick by templates generated from Blob picker. After 2D classification, 51,617 particles with different views were used for training in Topaz v0.2.4[70]. Subsequently, 1,739,854 total particles were extracted with a box size of 160 pixels and $2 \times 2$ binning and subjected to 2D classification, heterogeneous refinement, and Ab-Initio reconstruction. The well-defined subset of 242,089 particles was re-extracted with a box size of 224 pixels and refined to yield 4.47 Å map in non-uniform refinement with the settings of 12 Å initial low-pass resolution, minimization over per-particle scale enabled and five extra final passes. For further improving the density, seed-facilitated guided multi-reference 3D classification was performed[71]. The multi-references include accurate and biased references generated by Ab-Initio reconstruction, resolution gradient maps (4.47 Å map and other two maps with low-pass filtered resolution of 10 Å and 20 Å), and noise re-weighted maps (4.47 Å map and other two maps that micelle was downscaled by a factor of 0.3 and 0.7). After Ab-Initio reconstruction and removal of duplicated particles, 253,772 particles were re-extracted and subjected to heterogeneous refinement with only noise re-weighted maps, resulting in 3.84 Å. The resulted 145,869 particles and 242,089 particles from the initial processing were combined. Duplicates were removed after one round of heterogeneous refinement. After another two rounds of heterogeneous refinement, a total of 107,776 particles yielded a map at 3.72 Å. The final map was reconstructed (FSC = 0.143) at 3.59 Å for FtAlkB-nanobody or 3.45 Å for FtAlkB via Local refinement and Local CTF refinement. This map was sharpened respectively by DeepEMhancer[72] for better nanobody density and cryoSPARC for better ligand density. Local resolution was calculated by BlocRes in cryoSRARC.

### Model building and refinement

The AlphaFold2[53] FtAlkB model was fitted into the density in Chimera and manually rebuilt in Coot[73]. A nanobody model from PDB 7SL8 was fitted into the cryo-EM map, and the complementarity-determining regions (CDRs) were manually rebuilt with the assistance of Alpha-Fold2 prediction. The FtAlkB-nanobody model was refined in Phenix[74] and was evaluated by MolProbity. The refinement statistics reported in Supplementary Table 1 is based on the DeepEMhancer sharpend map. Figures were prepared using PyMOL (Schrödinger, LLC)[75], UCSF Chimera[76], or ChimeraX[77].

### Reporting summary

Further information on research design is available in the Nature Portfolio Reporting Summary linked to this article.

## Data availability

The data that support this study are available from the corresponding authors upon request. The FtAlkB-nanobody map has been deposited in the Electron Microscopy Data Bank (EMDB) under accession code EMD-40303 (FtAlkB-nanobody). The FtAlkB-nanobody model is in the Protein Data Bank (PDB) under 8SBB. Previously published structure models were downloaded from PDB using accession codes 6WF2, 4ZR1, 7SL8 Source data related to Figs. 1b, c, 3c, 4a, b, and Supplementary Fig. 5b are provided with this paper. Source data are provided with this paper.

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

## Acknowledgements

Thanks to Jacob Austin who helped develop the activity assay. Thanks to many students in the Austin group for fruitful discussions about AlkB

structure and function over the past 20 years. Thanks to Jamie Ross for help with ICP-MS data collection. S.W. and J.L. were recipients of Beckman Fellowships. A special acknowledgment to Professor Jay T. Groves, whose lab work on AlkB began as part of an NSF-funded center for Environmental Bioinorganic Chemistry (CEBIC NSF9810248) and whose intellectual engagement in this work has been invaluable. This work was made possible by support from NIH R01 GM130989 (R.N.A. & L.F.). Funds for purchasing the initial DNA used in the large-scale screen of AlkB came from a Presidential Research Award to R.N.A. from Barnard College. Some of this work was performed at the Stanford-SLAC Cryo-EM Center (S²C²) supported by the NIH Common Fund Transformative High-Resolution Cryo-Electron Microscopy program (U24 GM129541). Support for work in the Austin lab also comes from a generous gift from Roy and Diana Vagelos, which we also gratefully acknowledge.

## Author contributions

X.G. and J.L. performed initial screening and candidate verification. X.G., L.H., and J.Z carried out cryo-EM studies. X.G., L.H., and Y.X. performed ICP-MS. J.L., S.W., A.F., and R.N.A. performed functional studies. R.N.A. and L.F. directed the project. X.G., R.N.A., and L.F. wrote the manuscript.

## Competing interests

The authors declare no competing interest.
