## [Peer Review File · Nature Communications]

Reviewers' Comments:

Reviewer #1:

Remarks to the Author:

This manuscript reports the $\sim 3 \text{ \AA}$ resolution cryoEM structure of AlkB, an integral membrane iron monooxygenase that selectively hydroxylates the terminal methyl group of straight chain alkanes. AlkB enzymes play a key role in the global carbon cycle and have potential applications in biotechnology, which cannot be realized until their mechanism of activating inert hydrocarbon C-H bonds is understood. This structure has been a long time coming. The authors succeeded by screening an array of homologs for expression and stability and deploying state-of-the-art cryoEM to obtain a structure of the 45 kDa AlkB in complex with a stably-bound nanobody. The structure reveals the overall fold, the details of the alkane binding site, potential routes for substrate access and product egress, the molecular basis for substrate selectivity, and the molecular details of the diiron site. The structural results are beautifully presented and solidly supported by accompanying mutagenesis and activity assays. This work represents a significant step forward for the field.

Specific comments:

p. 3, line 25 and Fig. 1a: can the AlkB activity be given more quantitatively (i.e. amount of octanol per mg enzyme per time)? It would be useful to have a sentence with an exact number and then state how that number compares to the typical activity measured for the *P. oleovorans* AlkB.

p. 3, line 34: please specify here whether the AlkB for cryoEM is in a detergent or some other membrane-mimetic (methods say DDM, it would be helpful to state this here as well)

p. 4, line 7 and Fig. 1d: it is not clear where H2 and H3 are located. Are they labeled on Fig. 1d?

p. 4, line 17: a supplementary figure showing side by side cartoon structures of AlkB, SCD1, and Scs7p would help to emphasize the novelty of the AlkB fold. The statement here about the local similarity in the active sites is also mentioned starting on line 37 so could be removed from this part and just emphasized in the section below.

p. 4, line 30: ref. 47 indicates the presence of an antiferromagnetically coupled diiron center, which would have a shorter Fe-Fe distance than observed here. What exactly then is meant by "corroborates previous spectroscopic studies"?

p. 5, line 1 and Fig. 2a: what is meant by a "surrounding glutamate" in the Fig. 2a caption? Is the glutamate in coordinating distance of the Fe? Is it in hydrogen bonding distance of the nearby histidine ligand?

p. 5, line 32: please include Leu141 in Fig. 3b.

p. 7, line 3: maybe "ions" should be "irons"?

p. 7, para. 2: a figure showing the proposed mechanism should be included in the supplementary material.

Reviewer #2:

Remarks to the Author:

The authors present the cryoEM structure and functional characterisation of Alkane monooxygenase (AlkB), purified from the thermophilic bacterium *Fontimonas thermophila*. These data show that AlkB achieves catalysis via a di-iron active site that likely employs a distinct mechanism to that of another

hydrocarbon (methane) alkane oxidising enzyme sMMO. The structural also reveals a hydrophobic substrate binding channel, which provides alkane access to the di-iron active site. The authors demonstrate that the top of this substrate binding channel is gated by a tryptophan residue that plays a role in determining the length of the alkane that can be oxidised by AlkB.

The manuscript is well written and the results are clearly described. Determination of the structural a 45 kDa membrane protein is a considerable achievement and pushes the boundaries of what can be achieved using CryoEM. The functional analysis provided complements the structural data well and strengthens the conclusions drawn by the paper.

Considering the current accuracy of computational modelling of protein structure using Alphafold2 or an analogous method. Some of the claims in the paper are a bit overstated. For example, the authors state that 'it is unclear how/whether AlkB is related structurally to membrane desaturases and hydroxylases.' and 'the enzyme's structure is unknown.' I don't think these claims can be made considering the alphafold database contains a high quality model of AlkB (<https://alphafold.ebi.ac.uk/entry/P12691>). This is not to say that there isn't value in performing the experimental structural analysis described in the paper, as models require confirmation, and they don't provide details of substrate binding and active site architecture. However, I think it's better to be upfront about the fact that computational modelling now provides a very good indication of overall structure of most proteins. This is especially important considering the relatively low resolution of the CryoEM maps generated in this study. I note from the methods that the authors used Alphafold2 to generate their initial model but don't discuss this elsewhere in the manuscript. It would be useful to compare and contrast the similarities and differences between the experimental and computational structures.

In the discussion the authors speculate about the catalytic mechanism of AlkB, this is welcome, however, this discussion would be greatly assisted by a figure outlining the proposed mechanism.

Figure 3 would benefit from additional and more detailed views of the substrate binding channel and the proximity of the alkane density to the active site. It would be great if the authors could provide an animation of this region of the protein and the alkane density, as it's quite hard to judge the quality and location of the density from the images provided. The contour level for the density should also be provided.

Why do the authors think that the W62V mutant is worse at oxidising short chain alkanes, as well as being better at oxidising long ones? This question is driven by personal curiosity, but may be useful information to include in the manuscript.

Minor comments:

(Line numbers would be helpful for providing feedback)

Page 3 line 8 – 'Alkane-oxidising enzymes' not 'Alkane oxidising enzymes'

Page 7 line 10 – 'These observed comparable active site configuration from multiple distinct enzymes under different ...' there's a grammatical issue here, 'configurations' rather than 'configuration'?

Page 9 line 21 – 'Those 145,869 particles combined previous 242,089 particles ...' the meaning of this sentence isn't clear to me.

We thank the reviewers for providing constructive comments, which have helped us improve our manuscript. Below, we italicize referees' comments, then provide our responses.

Reviewer #1

This manuscript reports the ~3 Å resolution cryoEM structure of AlkB, an integral membrane iron monooxygenase that selectively hydroxylates the terminal methyl group of straight chain alkanes. AlkB enzymes play a key role in the global carbon cycle and have potential applications in biotechnology, which cannot be realized until their mechanism of activating inert hydrocarbon C-H bonds is understood. This structure has been a long time coming. The authors succeeded by screening an array of homologs for expression and stability and deploying state-of-the-art cryoEM to obtain a structure of the 45 kDa AlkB in complex with a stably-bound nanobody. The structure reveals the overall fold, the details of the alkane binding site, potential routes for substrate access and product egress, the molecular basis for substrate selectivity, and the molecular details of the diiron site. The structural results are beautifully presented and solidly supported by accompanying mutagenesis and activity assays. This work represents a significant step forward for the field.

We appreciate this reviewer's favorable summary assessment of our work and are delighted to be able to contribute to the field.

Specific comments:

p. 3, line 25 and Fig. 1a: can the AlkB activity be given more quantitatively (i.e. amount of octanol per mg enzyme per time)? It would be useful to have a sentence with an exact number and then state how that number compares to the typical activity measured for the P. oleovorans AlkB.

Yes, we routinely measure ~ 5 micromoles of product per mg of protein for FtAlkB in our assay. This is slightly higher than the optimized activity we have previously reported for the highly active AbAlkB (4.5 micromoles of product per mg of protein) (*Journal of Inorganic Biochemistry* 2013, 121: 46-52) and higher than what we typically measure for GPo1AlkB under similar conditions (2 micromoles of product per mg of protein). These numbers have been added to the paper (page 3, the 4th paragraph).

p. 3, line 34: please specify here whether the AlkB for cryoEM is in a detergent or some other membrane-mimetic (methods say DDM, it would be helpful to state this here as well)

We have added "purified in the detergent n-Dodecyl-β-D-Maltopyranoside (DDM)" in the main text (page 3, the 6th paragraph).

p. 4, line 7 and Fig. 1d: it is not clear where H2 and H3 are located. Are they labeled on Fig. 1d?

Thank you for pointing this out. We have now provided overall structures of FtAlkB in two orientations, and H2 and H3 are now labeled Fig. 1e.

p. 4, line 17: a supplementary figure showing side by side cartoon structures of AlkB, SCD1, and Scs7p would help to emphasize the novelty of the AlkB fold. The statement here about

the local similarity in the active sites is also mentioned starting on line 37 so could be removed from this part and just emphasized in the section below.

We thank the reviewer for pointing this out. We have included side by side comparisons between AlkB, SCD1 and Scs7p in Supplementary Fig. 4e-g. The redundant statement here on the active site has been removed.

p. 4, line 30: ref. 47 indicates the presence of an antiferromagnetically coupled diiron center, which would have a shorter Fe-Fe distance than observed here. What exactly then is meant by “corroborates previous spectroscopic studies”?

We agree with the reviewer. Our intended meaning is that the nitrogen-containing ligand identity is confirmed. We have revised the sentence as follows: “This corroborates the identity of metal ligands surmised from previous spectroscopic studies of AlkB⁵¹”

p. 5, line 1 and Fig. 2a: what is meant by a “surrounding glutamate” in the Fig. 2a caption? Is the glutamate in coordinating distance of the Fe? Is it in hydrogen bonding distance of the nearby histidine ligand?

In our cryoEM map, the local density of this glutamate’s side chain is relatively weak, which is probably due to the increased sensitivity of negatively charged residues to radiation damage during cryo-EM data collection (this is a relatively commonly observed phenomenon in EM maps). As a result, its side chain rotamer cannot be reliably assigned. Nonetheless, if E281 assumes the most populated rotamer, its side chain oxygens would be positioned about 3.6 Å to Fe1 and within hydrogen bond distance to adjacent histidine residues. We have clarified this point in the text, modifying the sentence to say “Nevertheless, there are important differences: there is a carboxyl group of a conserved glutamate residue, E281, in AlkB that is near Fe1 (the iron coordinated by four histidine residues) that could potentially interact with that iron and nearby histidine residues, although the side chain of E281 is not sufficiently resolved in the cryo-EM map.” (page 5, 2nd paragraph).

p. 5, line 32: please include Leu141 in Fig. 3b.

We have followed the advice and added L141 in Fig. 3b.

p. 7, line 3: maybe “ions” should be “irons”?

We have edited it to “iron ions”.

p. 7, para. 2: a figure showing the proposed mechanism should be included in the supplementary material.

We have followed the advice and included a figure showing the proposed mechanism in Supplementary Fig. 6.

Reviewer #2

The authors present the cryoEM structure and functional characterisation of Alkane monooxygenase (AlkB), purified from the thermophilic bacterium Fontimonas thermophila.

These data show that AlkB achieves catalysis via a di-iron active site that likely employs a distinct mechanism to that of another hydrocarbon (methane) alkane oxidising enzyme sMMO. The structural also reveals a hydrophobic substrate binding channel, which provides alkane access to the di-iron active site. The authors demonstrate that the top of this substrate binding channel is gated by a tryptophan residue that plays a role in determining the length of the alkane that can be oxidised by AlkB.

The manuscript is well written and the results are clearly described. Determination of the structural a 45 kDa membrane protein is a considerable achievement and pushes the boundaries of what can be achieved using CryoEM. The functional analysis provided complements the structural data well and strengthens the conclusions drawn by the paper.

We appreciate the reviewer's favorable summary assessment of our work.

Considering the current accuracy of computational modelling of protein structure using AlphaFold2 or an analogous method. Some of the claims in the paper are a bit overstated. For example, the authors state that 'it is unclear how/whether AlkB is related structurally to membrane desaturases and hydroxylases.' and 'the enzyme's structure is unknown.' I don't think these claims can be made considering the alphafold database contains a high quality model of AlkB (<https://alphafold.ebi.ac.uk/entry/P12691>). This is not to say that there isn't value in performing the experimental structural analysis described in the paper, as models require confirmation, and they don't provide details of substrate binding and active site architecture. However, I think it's better to be upfront about the fact that computational modelling now provides a very good indication of overall structure of most proteins. This is especially important considering the relatively low resolution of the CryoEM maps generated in this study. I note from the methods that the authors used AlphaFold2 to generate their initial model but don't discuss this elsewhere in the manuscript. It would be useful to compare and contrast the similarities and differences between the experimental and computational structures.

We, and others, are still learning about the predictive capacities of AlphaFold, especially for metalloenzymes where metal ions have the potential to impact the structure. We have taken the advice and revised the statement. Now we focus on the point that the exact configuration of AlkB's diiron center remains unclear (page 2, the 4th paragraph) and that its structure is yet to be experimentally determined (page 3, the 1st paragraph). Moreover, we have now provided an overlay of the AlphaFold predicted structure and our experimental structure in Supplementary Fig. 4e, as well as a discussion about the effectiveness and limitation of the AlphaFold-predicted structure of AlkB (page 6, the 4th paragraph).

In the discussion the authors speculate about the catalytic mechanism of AlkB, this is welcome, however, this discussion would be greatly assisted by a figure outlining the proposed mechanism.

Thank you for the advice. We have now included a figure of the proposed mechanism in Supplementary Fig. 6.

Figure 3 would benefit from additional and more detailed views of the substrate binding channel and the proximity of the alkane density to the active site. It would be great if the authors could provide an animation of this region of the protein and the alkane density, as it's quite hard to judge the quality and location of the density from the images provided. The

contour level for the density should also be provided.

We have now added a panel in Supplementary Fig. 4d to show additional views of the substrate binding channel. Moreover, we have now provided the density of ligand, with contour level, in two different views in Supplementary Fig. 4c.

Why do the authors think that the W62V mutant is worse at oxidising short chain alkanes, as well as being better at oxidising long ones? This question is driven by personal curiosity, but may be useful information to include in the manuscript.

We think this is because the non-polar alkane needs to be held in place by the bulky amino acid and when this is mutated to a less bulky amino acid, shorter alkanes have considerably more mobility in the active site, which in turns leads to less efficient oxidation. This is a point we discussed in some detail in prior work (*Angew. Chemie. Int. Ed., Engl.* 2006, 45: 8192-8194).

We now add a sentence “It is conceivable that the bulky amino acid may serve as a barrier to longer alkanes while also helping to hold optimally sized non-polar alkanes in place” to the discussion along with a reference to the prior work (page 6, the 5th paragraph).

Minor comments:

(Line numbers would be helpful for providing feedback)

Page 3 line 8 – ‘Alkane-oxidising enzymes’ not ‘Alkane oxidising enzymes’

Corrected.

Page 7 line 10 – ‘These observed comparable active site configuration from multiple distinct enzymes under different ...’ there’s a grammatical issue here, ‘configurations’ rather than ‘configuration’?

Thank you, this has been corrected.

Page 9 line 21 – ‘Those 145,869 particles combined previous 242,089 particles ...’ the meaning of this sentence isn’t clear to me.

We thank the reviewer for pointing it out. We have now edited this sentence to “The resulted 145,869 particles and 242,089 particles from the initial processing were combined. Duplicates were removed after one round of heterogeneous refinement.” for clarification.

Overall, the manuscript has undergone one more round of editing for improved grammar and clarity.